# Impact of Face Mask-Wearing on Quality of Life in Post-Surgical Oral Cancer Patients: A Cross-Sectional Study

**DOI:** 10.3390/cancers16244199

**Published:** 2024-12-17

**Authors:** Romain Lan, Frédéric Silvestri, Maryem Rhanoui, Cassandre Bezier, Nicolas Fakhry, Florence Carrouel, Chloé Mense

**Affiliations:** 1Aix-Marseille Univ CNRS, ADES, 13015 Marseille, France; 2Functional Unit of Maxillo-Facial Prosthesis, Timone University Hospital, 13005 Marseille, France; 3Laboratory “Health, Systemic, Process” (P2S), UR4129, University Claude Bernard Lyon 1, University of Lyon, 69008 Lyon, France; 4Otorhinolaryngology-Head and Neck Surgery Department, La Conception University Hospital, 13005 Marseille, France

**Keywords:** oral squamous cell carcinomas, mask-wearing, quality of life

## Abstract

Oral cancer remains one of the most common cancers globally, with invasive treatment often resulting in significant disfigurement and psychosocial distress. During the COVID-19 pandemic, the mandatory mask-wearing policies offered a unique opportunity to assess the impact of facial coverings on the quality of life (QoL) of these patients. This study provides valuable insights into how a simple intervention—mask-wearing—can positively influence the well-being of patients. It enhanced the QoL of oral cancer patients, reduced social anxiety and improved self-perception, indicating the dual role of masks as both a health protective measure and a psychosocial tool.

## 1. Introduction

Oral squamous cell carcinomas (OSCCs) are the sixth most common cancer worldwide, with a five-year survival rate around 50%, posing a major public health challenge despite advances in diagnostics and therapeutics [1,2]. Extensive surgical resection, often associated with radiotherapy, remains the optimal first-line treatment for OSCCs, but it can result in disfigurement, altered speech, and swallowing difficulties, exacerbating patients’ psychosocial burden [3,4,5]. This aesthetic and functional problems cause significant emotional distress, impair social interactions and communication, and can lead to anxiety or depression, severely affecting patient quality of life (QoL) [6,7,8].

In the context of the COVID-19 pandemic, governments worldwide implemented mandatory mask-wearing policies in public settings to mitigate viral transmission (World Health Organization, 2020). While masks are often seen as uncomfortable and socially inhibiting by the general population, for patients treated for OSCC, the widespread acceptance of mask-wearing during the pandemic might provide psychological and social benefits by normalizing facial differences and reducing stigma, thus promoting inclusivity and acceptance [9,10].

However, wearing a mask can also be problematic for these patients, as it may cause physical discomfort and functional limitations, exacerbate sensory deficits or mucosal irritation, and increase airway resistance and breathing difficulties in those with respiratory or swallowing difficulties [5,11].

Therefore, due to a lack of evidence, assessing OSCC patients’ experiences and concerns regarding mask-wearing and their impact on QoL is crucial to develop tailored supportive care interventions. This study aimed to analyze the impact of mask-wearing on QoL in patients surgically treated for OSCC.

## 2. Materials and Methods

### 2.1. Study Design and Setting

This single-center, cross-sectional, descriptive study was carried out at Timone Hospital, Marseille, France, between June 2022 and December 2023. The study was conducted in accordance with the principles of the Declaration of Helsinki and approved by the Ethics Committee of the University of Aix-Marseille (N/ 2021-11-018-03—n° 48GJJB). This study was reported in accordance with the “Strengthening the reporting of observational studies in epidemiology (STROBE)” checklist (Appendix A) [12].

### 2.2. Study Justification

During the COVID-19 pandemic, in France, the obligation to wear a mask on a daily basis was introduced on 11 May and abolished on 14 March 2022 (French Government, 2022). The announcement of the end of the obligation led many patients surgically treated for oral cancer to express their apprehension about removing the mask outdoors. This led to the development of a study to assess the impact of wearing a mask on the QoL of these patients.

### 2.3. Participants and Study Process

The target population was patients received in consultation for advice on maxillofacial prosthetic rehabilitation following carcinologic treatment. A practitioner presented and described the study to the patient. After agreeing to participate in the study, the patient signed an informed consent form and anonymously completed the QoL questionnaire.

### 2.4. Inclusion and Exclusion Criteria

To be included, the patients had to (i) be over 18 years of age; (ii) have undergone OSCC surgical treatment; (iii) have returned home after OSCC treatment before 14 March 2022 (the end of mandatory mask-wearing in France).

Patients were excluded if they (i) did not accept to answer the questionnaire; (ii) were unable to answer the questionnaire (due to a physical or mental disability or the inability to speak French); (iii) were treated for OSCC after the pandemic period.

### 2.5. Outcomes

The primary outcome was the evaluation of the impact of face mask-wearing on the QoL of patients surgically treated for OSCC.

The secondary outcome was the evaluation of the influence on the QoL of patients surgically treated for OSCC of (i) physical appearance; (ii) initial tumor localization; (iii) radiotherapy; (iv) end date of cancer treatment.

### 2.6. Sample Size

The sample size calculation, based on the primary outcome and on previous studies assessing QoL in similar patient populations, suggested a moderate effect size (d = 0.6) [13,14,15]. Using the formula for estimating the sample size in cross-sectional studies with continuous outcomes, assuming a significance level (alpha) of 0.05 and a statistical power of 0.80 and estimating the standard deviation of the QoL scores to be 1.0., the required sample size was calculated to be approximately 39 patients.

### 2.7. Questionnaire Design

A specific questionnaire on the impact of wearing a mask was developed (Appendix A) by selecting, without modifications, 9 questions (Q) from the Adult Strabismus-20 questionnaire and the Orthognathic Quality of Life Questionnaire [15,16]. The selected questions targeted 9 self-perception factors: appearance (Q1), fear of family’s perception (Q2), fear of others’ perception (Q3), fear of being photographed (Q4), fear of others’ judgment (Q5), fear of hurtful comments (Q6), self-confidence (Q7), appearance-related depression (Q8), and sociability apprehension (Q9).

For each question, the patients were asked to indicate their feelings on a four-point Likert scale for two conditions: “with mask” and “without mask”.

Each question was scored from 1 point (1, not at all) to 4 points (4, a lot). The mean of the score of each question permitted to obtain the overall score. The higher the overall score was, the lower the QoL.

### 2.8. Assessment of Internal Consistency and Score Distribution

To evaluate the validity and the reliability of the questionnaire, Cronbach’s Alpha was calculated to measure the internal consistency of the questionnaire items, in the conditions with and without mask use, separately. This analysis permitted to determine the extent to which the questionnaire items measured the same concept. A high Cronbach’s alpha indicated that the items within the scale were well correlated, suggesting that the scale was reliable. The interpretation of the Cronbach’s alpha coefficient was as follows: excellent (≥0.90), good (0.80–0.89), acceptable (0.70–0.79), questionable (0.60–0.69), poor (0.50–0.59), and unacceptable (<0.50) [16].

The distribution of the overall scores for each question was evaluated using the Shapiro–Wilk test to indicate the normality or lack of it of the distributions. Floor or ceiling effects were calculated and considered as present when more than 15% of the responses were rated with the minimum or maximum scores, respectively.

### 2.9. Data Analysis

For each participant, the responses were classified according to four categories, each comprising two modalities: (i) physical appearance (altered versus unaltered); (ii) initial tumor localization (maxillary versus mandibular); (iii) radiotherapy (yes versus no); (iv) date of the end of the treatment (return home before versus during the COVID-19 pandemic).

For physical appearance, altered or unaltered physical appearance was defined as the general morphology with or without visible hard and/or soft tissue lesions on the face [17].

For the date of the end of treatment, patients operated before March 2020 and who had experienced a social life without compulsory mask-wearing were classified as having an “end of treatment before the COVID-19 pandemic”. Conversely, patients operated between March 2020 and March 2022, who first experienced a social life with mandatory mask-wearing before the restrictions were lifted, were classified as having an “end of treatment during the COVID-19 pandemic”.

### 2.10. Statistical Analysis

Qualitative variables are reported in absolute values (percentage, %). Quantitative variables are reported as means with standard deviation (SD).

The participants in the two modalities of the same category subgroups (e.g., (i) patients with altered physical appearance and (ii) patients with non-altered physical appearance) were distinct from one another, while the participants in the same modality (e.g., with and without a mask) were the same individuals, ensuring paired comparisons for within-modality analyses.

Paired *t*-tests were used for comparing the differences within each group to the main outcome.

For category-specific analyses (physical appearance, initial tumor localization, radiotherapy, and end date of treatment), the Wilcoxon signed-rank test was used to compare differences in the responses of patients in the “with” and “without mask” conditions within each subgroup, accounting for potential differences in sample size and non-normal distributions. This test was also applied to evaluate interaction effects on the overall mean scores across subgroups. The differences based on the overall mean scores of the two independent subgroups (e.g., patients with altered physical appearance vs. patients with non-altered physical appearance) with different variances were analyzed with Welch’s *t*-test.

To evaluate correlations between categories, the categorical variables were converted into numerical values using label encoding, and a Pearson correlation matrix was computed to measure the relations between categories and response means.

To evaluate the influence of the categories on the responses to questions according to the mask condition, a mixed-effects regression model was employed. This approach was well-suited as it accounts for both fixed and random effects. The fixed effects were the systematic influences of predictors such as physical appearance, reaction time, localization, and COVID-19 period on the responses. These effects were assumed to be consistent across the participants. The random effects captured variability at the individual level, accounting for differences in how the participants responded due to intrinsic or unmeasured factors.

A *p*-value ≤ 0.05 was considered statistically significant.

The statistical analysis was carried out using Python 3.10, with statsmodels.regression.mixed_linear_model for the mixed-effects regression model, and sklearn.preprocessing for data preprocessing and encoding. Statistical calculations were performed using scipy.stats, while matplotlib and seaborn were used for data visualization.

## 3. Results

### 3.1. Characteristics of the Patients

The characteristics of the 41 patients included are summarized in Table 1. The median age was 69 ± 8.4 years. The majority of the patients were male (63%).

Twenty-seven patients (66%) exhibited an alteration in their physical appearance. The initial tumor was primarily located in the mandibular region for 23 patients (56%). Additionally, 34 patients (83%) were treated with radiotherapy, and 25 patients (61%) underwent surgery during the COVID-19 pandemic.

### 3.2. Assessing the Questionnaire Validity Through Internal Consistency and Score Distribution Analysis

The descriptive analysis of the questionnaire is presented in Appendix A. High ceiling and floor effects were observed, suggesting variability in the responses between patients and suggesting that the mask condition significantly affected the QoL. Shapiro–Wilk’s test confirmed these results and demonstrated a non-normal distribution for all the questions.

However, the calculated Cronbach’s alpha was approximately 0.931, indicating a high level of internal consistency among the questions (“with” and “without mask”, separately).

### 3.3. Impact of Face-Mask Wearing on the QoL

The overall mean scores for the “with mask” and “without mask” conditions were statistically different (*p* < 0.001) and smaller for patients who wore a mask (1.66 vs. 2.00 for the “without mask” condition, Table 2 and Figure 1A). Therefore, the QoL of OSCC patients was significantly better when they wore a mask.

For all nine questions, the score was higher for the “without mask” than for the “with mask” condition, corresponding to a lower QoL for patients in the “without mask” condition. This difference was only significant for questions about appearance (Q1, *p* = 0.008), fear of family’s perception (Q2, *p* = 0.003), fear of others’ perception (Q3, *p* < 0.001), fear of being photographed (Q4, *p* = 0.015), and sociability apprehension (Q9, *p* < 0.001) (Figure 1B).

### 3.4. Impact of Physical Appearance, Initial Tumor Localization, Radiotherapy, and Date of the End of Treatment on QoL, with and Without the Use of a Mask

Wearing a mask was associated with significant lower overall mean scores for all the categories (physical appearance, initial tumor localization, radiotherapy, and date of the end of treatment), indicating that these categories had significant impact on patients’ QoL as indicated for the overall *p*-value by the Wilcoxon signed-rank test (Table 3).

Concerning the score distribution within each category (Figure 2), the QoL was significatively better with the use of a mask for the patients, whatever the initial tumor localization and end of treatment period; this was also observed for patients with altered physical appearance and who received radiotherapy.

Regarding the two mask conditions separately, QoL was better for patients with non-altered physical appearance (significantly) and mandibular localization (not significantly), with and without mask use, as indicated by Welch’s *t*-test (Table 3 and Appendix A). Conversely, the overall mean scores were higher, but not significantly, for patients without radiotherapy and those whose treatment ended before the COVID-19 pandemic, whatever the mask condition.

The relationships between the categories based on the Pearson correlation matrix revealed a moderate positive correlation between physical appearance and initial tumor localization (r = 0.40), suggesting that specific tumor sites are more likely to alter the physical appearance. Weak correlations among other variables (∣r∣ < 0.23) indicated minimal multicollinearity, supporting the independence of the predictors in the mixed-effects model (Figure 3).

### 3.5. Analysis of the Association Between the Nine Self-Perception Factors and Physical Appearance, Initial Tumor Localization, Radiotherapy, and Date of the End of Treatment

The analysis of the relationship, with or without a mask, between the nine self-perception factors and physical appearance, initial tumor localization, radiotherapy, and date of the end of treatment revealed both positive and negative impacts of these predictors (Figure 4). Altered physical appearance consistently demonstrated a notable positive influence on patient-reported outcomes across all questions, regardless of mask use, highlighting its persistent impact on perceived experiences and concerns. Maxillary tumor localization showed a generally positive but weaker association across most questions in both mask conditions. However, a strong positive association was observed specifically for Q3 (fear of others’ perception), Q5 (fear of others’ judgment), and Q6 (fear of hurtful comments). These results emphasize the critical role of tumor site, particularly maxillary localization, in shaping patient responses, potentially due to its functional or esthetic impact, which may be more pronounced under the “with mask” condition.

In contrast, the generally negative coefficients associated with the end of treatment during the COVID-19 pandemic and radiotherapy suggested a potential dampening effect on patient experiences. This was particularly evident for questions related to comfort and interpersonal interactions, such as Q6 and Q7, as the pandemic period and radiotherapy might have amplified psychological or social challenges. These findings underline the importance of contextual factors and treatment-related variables in influencing self-perception outcomes.

## 4. Discussion

To our knowledge, this is the first study to assess the impact of wearing a mask on the QoL of patients who were surgically treated for OSCC between the beginning and the end of the restrictions due to the COVID-19 pandemic.

The validity of the measures was inferred from Cronbach’s alpha values, which indicated an excellent internal consistency. This internal consistency across the questions indicated that the questions were appropriately correlated.

OSCC patient QoL was better when wearing a mask compared to when not wearing a mask (overall mean scores, respectively, of 2 and 2.88, *p* < 0.001, Table 2). This improvement was especially noticeable for questions concerning patient perception of appearance (*p* = 0.008), fear of family’s perception (*p* = 0.003), fear of others’ perception (*p* < 0.001)), fear of being photographed (*p* = 0.015), and sociability apprehension (*p* < 0.001).

These questions all targeted the psychosocial theme of appearance, which may seem consistent with the management of these cancers, which modify the patient’s individual and social identity [18,19]. Studies on the psychological effects of carcinological surgery for these cancers show that one of the main problems is linked to physical appearance, which is the cause of significant psychological and social distress for patients, with the face playing a predominant role in social relationships [20,21,22,23]. Moreover, a significant impact was observed regarding how others perceived patients who had re-engaged in social activities post-operation, first with the mask on and then without it. This highlights the crucial psychological and social role that the mask plays for these patients [24,25].

These results underline the importance of considering both physical and contextual factors in interpreting patient outcomes with and without the use of a mask.

Whatever the initial tumor localization (maxillary or mandibular), the absence of a mask reduced the QoL of OSCC patients due to the fear of other people’s views, reflecting the physical and psychosocial impact of these cancers and their treatment [26,27].

However, wearing a mask had a greater impact on patients suffering from mandibular lesions in terms of their appearance and the way they are viewed by those around them.

This can be explained by the fact that, in our study, the majority of the patients with mandibular damage showed physical alterations, whereas less than half of the patients with maxillary damage showed physical sequelae. The loss of mandibular continuity can lead to significant aesthetic deformity due to the loss of the lower contour of the face [17]. This may also be due to the carcinologic localization, which is mainly on the hard or soft palate for the maxilla, and therefore with less aesthetic or cutaneous impact.

More predictably, the patients whose physical appearance was not altered did not seem to be affected by wearing a mask. For them, the mask appeared to be a simple sanitary barrier causing potential undesirable effects. Indeed, a study by Kanzow et al. demonstrated that the mask increased the perception of dry mouth and halitosis [28]. For patients with altered physical appearance, this study’s results are similar to those of Dupuy and Satabin, who evaluated the impact of wearing a mask on QoL in patients with peripheral facial paralysis [29]. The mask positively influenced these patients psychologically, as well as functionally and socially, improving communication and the transmission of emotions. More than a health barrier, the mask acted as “social safety net”, a finding reinforced by our study, where mask-wearing significantly positively impacted on psychosocial well-being.

The patients’ QoL was better whether they were treated before the COVID-19 pandemic and returned to social life without a mask or during the pandemic with compulsory mask-wearing. The invasiveness of treatments and their aesthetic, functional, and psychological consequences had a greater impact than the visual protection offered by the mask.

However, this study has several limitations. The small sample size limits the interpretation of the results between categories and comparisons with other larger scale studies [28,29]. The sample selected had to be correlated with epidemiological data on cancers of the upper aerodigestive tract, the smaller number of patients requiring maxillofacial prosthesis treatment, and the short inclusion period due to the end of the mask-wearing obligation. This sample size limitation may explain the lack of significant findings. This could particularly account for the results related to radiotherapy, where it appears, from this very small sample (n = 7 for the subcategory without radiotherapy), that the patients who did not receive radiotherapy had a higher quality of life (both with and without a mask) compared to those who did. These results and differences in sample size may also be explained by recruitment biases related to local surgical and oncological practices. The gold standard for treating patients with OSCC is surgery, and when medically feasible, it is combined with radiotherapy [30].

Additionally, the variability and multiplicity of the carcinologic lesions did not allow for a homogenous distribution of the localizations (maxillary/mandibular) [31]. While the modest sample size and the specific context of the COVID-19 pandemic may limit the generalizability of our findings, this study serves as a valuable proof of concept, demonstrating the potential of mask-wearing as a simple, low-cost intervention to enhance QoL for OSCC patients. Future larger, multicenter studies are warranted to validate these results and further explore the integration of such measures into routine supportive care strategies for cancer patients.

For all these reasons, these results and their interpretation should be approached with extreme caution, and no definitive conclusions should be drawn.

## 5. Conclusions

Wearing a mask seemed to have a positive impact on the QoL of patients with physical impairment resulting from OSCC. In addition to its health benefits, a mask in everyday practice, being easily accessible and simple to use, could provide psychological and social benefits, constituting a waiting solution before definitive prosthetic treatment.

By analyzing feelings from the patients’ perspective, this study highlights the complex interactions between mask use, psychosocial well-being, and functional outcomes in this vulnerable population. Given the significant impact of facial aftereffects on quality of life and the risk of respiratory infection, continuing to wear a mask could be beneficial. This consideration could inform tailored interventions and supportive care strategies, optimizing patients’ adjustment and adaptation following post-cancer surgery.

## Figures and Tables

**Figure 1 cancers-16-04199-f001:**
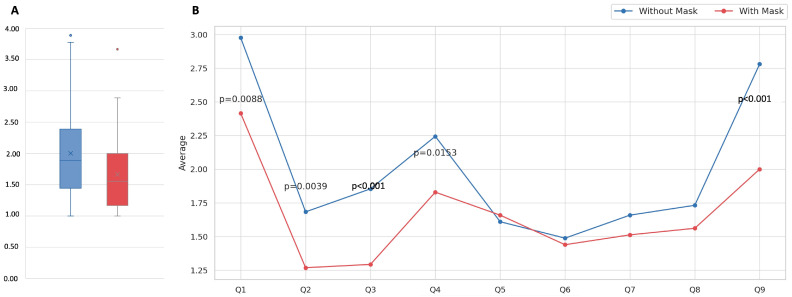
(**A**). Box plot of overall mean scores for the “with” and “without mask” conditions; (**B**). line plot of mean scores for each question, with significant *p*-values for Q1, Q2, Q3, Q4, and Q9.

**Figure 2 cancers-16-04199-f002:**
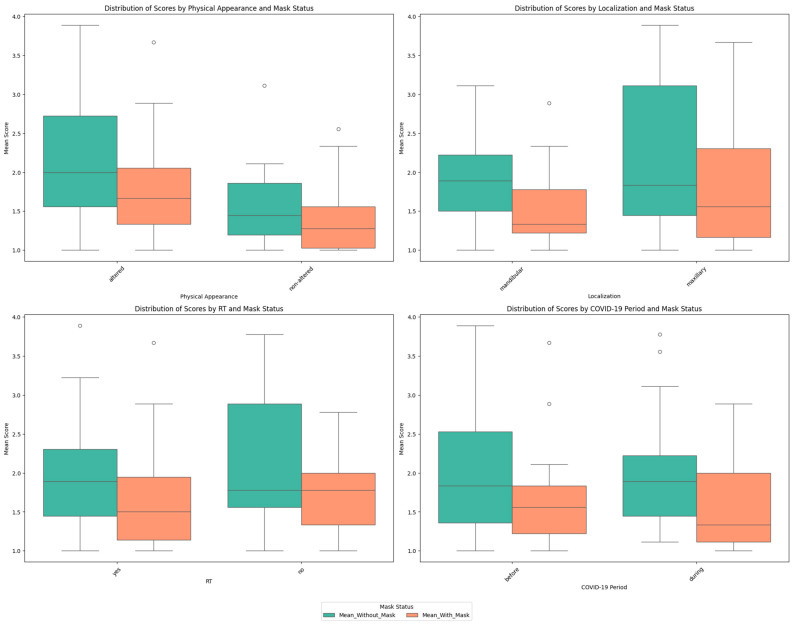
Box plot distribution of scores for the “with” and “without mask” conditions according to physical appearance, initial tumor location, radiotherapy, and date of end of treatment.

**Figure 3 cancers-16-04199-f003:**
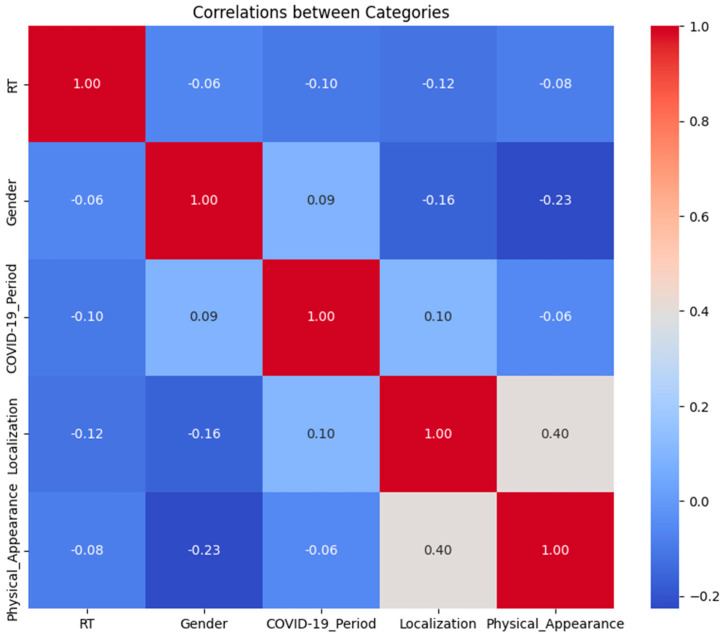
Heatmap of the correlation based on the mixed-effects model regression analyzing the relationships between the categories. The color intensity represents the magnitude and direction of the coefficients (blue for negative, and red for positive).

**Figure 4 cancers-16-04199-f004:**
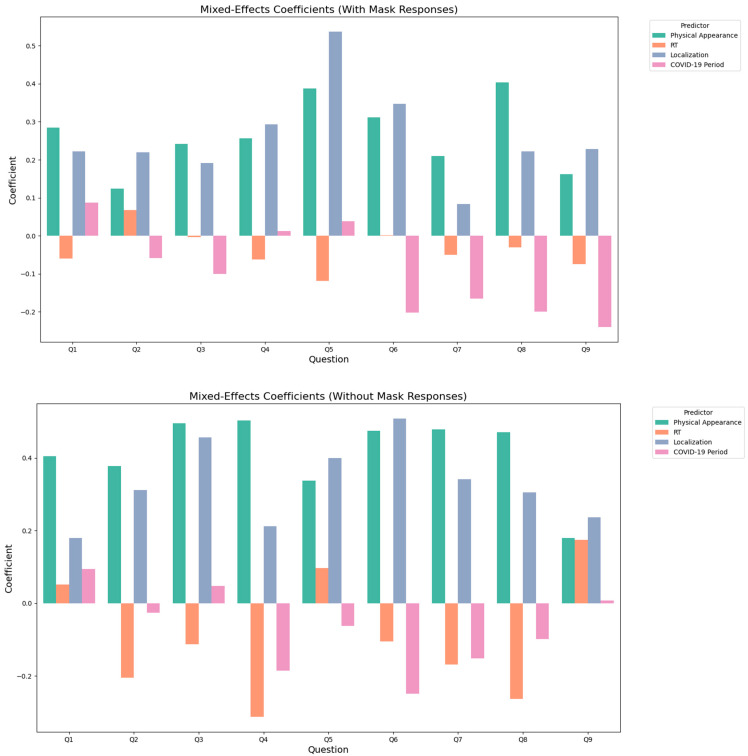
Bar plots of mixed-effects model coefficients illustrating the impact of categorical predictors on the responses to the 9 self-perception questions, separated by mask condition (“without mask” and “with mask”). Each bar represents the coefficient’s magnitude and direction, with positive values indicating an increase in the response score, and negative values indicating a decrease.

**Table 1 cancers-16-04199-t001:** Population characteristics (n = 41).

Variables		n %
Age (mean ± SD)		69 ± 8.4
Gender		
	Male	26 (63%)
	Female	15 (37%)
Physical Appearance		
	Altered	27 (66%)
	Non altered	14 (34%)
Initial tumor localization		
	Maxillary	18 (44%)
	Mandibular	23 (56%)
Radiotherapy		
	Yes	34 (83%)
	No	7 (17%)
Date of the end of treatment		
	Before COVID-19 pandemic	16 (39%)
	During COVID-19 pandemic	25 (61%)

**Table 2 cancers-16-04199-t002:** Comparison of mean scores overall and by question for patients wearing and not wearing a mask.

Question	Mean Score (SD)		*p*
	With Mask	Without Mask	
Q1. Appearance	2.41 (1.22)	2.98 (1.01)	**0.008**
Q2. Fear of family’s perception	1.27 (0.63)	1.68 (0.93)	**0.003**
Q3. Fear of others’ perception	1.29 (0.68)	1.85 (1.04)	**<0.001**
Q4. Fear of being photographed	1.83 (1.12)	2.24 (1.28)	**0.015**
Q5. Fear of others’ judgment	1.66 (1.15)	1.61 (0.97)	0.7
Q6. Fear of hurtful comments	1.44 (0.81)	1.49 (0.95)	0.623
Q7. Self-confidence	1.51 (0.78)	1.66 (0.88)	0.294
Q8. Appearance-related depression	1.56 (0.98)	1.73 (1.03)	0.241
Q9. Sociability apprehension	2.00 (1.16)	2.78 (1.13)	**<0.001**
Overall	1.66 (1.02)	2.00 (1.14)	**<0.001**

Bold highlights results for which there is a significant difference.

**Table 3 cancers-16-04199-t003:** Overall mean score comparison according to physical appearance, initial tumor localization, radiotherapy, and date of the end of treatment. ^a^: Wilcoxon signed-rank test; ^b^: Welch’s *t*-test.

Category	Subgroup	Without Mask	With Mask	*p*-Value ^a^	Overall *p*-Value ^a^ (With and Without Mask)
Physical appearance	Altered (n = 27)	2.20 (0.82)	1.77 (0.67)	**<0.001**	**<0.001**
	Non altered (n = 14)	1.63 (0.57)	1.45 (0.51)	0.13	
	*p*-value ^b^	**0.01**	**0.01**		
Initial tumor localization	Maxillary (n = 18)	2.19 (0.97)	1.83 (0.78)	**0.01**	**<0.001**
	Mandibular (n = 23)	1.86 (0.58)	1.53 (0.46)	**<0.001**	
	*p*-value ^b^	0.2	0.16		
Radiotherapy	Yes (n = 34)	1.96 (0.73)	1.65 (0.64)	**<0.001**	**<0.001**
	No (n = 7)	2.21 (1.06)	1.75 (0.61)	0.06	
	*p*-value ^b^	0.58	0.71		
End of treatment	Before COVID-19 pandemic (n = 16)	2.01 (0.91)	1.72 (0.71)	**0.01**	**<0.001**
	During COVID-19 pandemic (n = 25)	2.00 (0.72)	1.63 (0.59)	**<0.001**	
	*p*-value ^b^	0.98	0.66		

Bold highlights results for which there is a significant difference.

## Data Availability

The original contributions presented in this study are included in the article/Appendix A. Further inquiries can be directed to the corresponding author(s).

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
