# Peer review of "Impact of Face Mask-Wearing on Quality of Life in Post-Surgical Oral Cancer Patients: A Cross-Sectional Study"

_cancers, 2024, doi:10.3390/cancers16244199_

Round 1

Reviewer 1 Report

Comments and Suggestions for Authors

The manuscript reports the results of a study conducted through the use of a questionnaire administered to patients with oral squamous cell carcinoma. This questionnaire aimed to evaluate the effect of face masks on the quality of life perceived by the interviewees. After a careful statistical evaluation of the answers provided by the 41 patients involved in the study, the authors arrive at the conclusion that the patients who were most dissatisfied with their appearance due to the pathology and any interventions they had undergone (surgery or radiotherapy), had gained some psychological benefit from the use of face masks.

I do not believe that the research described in this manuscript falls within the scope of the journal. Furthermore, the study methods reported were not used appropriately. n fact, given what the authors state in the first sentence of the introduction (Oral squamous cell carcinomas (OSCCs) are the sixth most common cancer worldwide) and taking into account that the inclusion criteria used for patient selection were very loose, a cohort of only 41 patients is absolutely undersized.

Therefore, I do not consider this manuscript suitable for publication in Cancer.

Author Response

Comment 1 : The manuscript reports the results of a study conducted through the use of a questionnaire administered to patients with oral squamous cell carcinoma. This questionnaire aimed to evaluate the effect of face masks on the quality of life perceived by the interviewees. After a careful statistical evaluation of the answers provided by the 41 patients involved in the study, the authors arrive at the conclusion that the patients who were most dissatisfied with their appearance due to the pathology and any interventions they had undergone (surgery or radiotherapy), had gained some psychological benefit from the use of face masks.

I do not believe that the research described in this manuscript falls within the scope of the journal.

Response 1 : Thank you for your feedback and for taking the time to review our manuscript. We would like to address your concern regarding whether our research falls within the scope of Cancers.

  1. Link between quality of life (QoL) and oncology: Cancers is a journal that addresses the multidimensional aspects of cancer, including post-treatment quality of life. Our study focuses on a crucial aspect of supportive care in oncology: the psychological impact of treatments and how patients cope with the aesthetic consequences of surgical and radiotherapeutic interventions. Although face masks were primarily seen as a public health measure during the pandemic, our research reveals their potential as a psychological aid for some cancer patients. This dimension is directly aligned with holistic cancer care, which is a core theme of Cancers.
  2. Contribution to supportive care strategies: Supportive care is a vital component of oncology, especially for patients with oral squamous cell carcinoma (OSCC), where aesthetic disfigurement can significantly impair quality of life. Our study introduces a simple intervention—mask-wearing—that has a positive impact on patients’ QoL. This is an innovative finding that addresses a clinical need, as few studies explore practical ways to improve patients’ body image and psychosocial well-being during the post-treatment phase.
  3. Relevance in the post-pandemic context: The COVID-19 pandemic introduced new dynamics in patient care, and mask-wearing had unexpected benefits for patients with facial disfigurements. Our study demonstrates that this practice can be integrated into supportive care strategies for OSCC patients, particularly in the post-pandemic context. As such, our research explores a novel and relevant dimension of cancer patient management, which we believe is important for a journal like Cancers to address.
  4. Promoting comprehensive care for OSCC patients: Cancer treatment extends beyond medical interventions; it involves addressing the psychosocial and aesthetic repercussions of those treatments, which is especially pertinent for OSCC patients. In this sense, our manuscript contributes to the discussion on the overall impact of cancer treatments, a crucial issue that aligns with the broader scope of Cancers.

We hope that these clarifications demonstrate how our study contributes to the journal's focus and adds value to the conversation on the comprehensive care of cancer patients.

Comment 2 : Furthermore, the study methods reported were not used appropriately. In fact, given what the authors state in the first sentence of the introduction (Oral squamous cell carcinomas (OSCCs) are the sixth most common cancer worldwide) and taking into account that the inclusion criteria used for patient selection were very loose, a cohort of only 41 patients is absolutely undersized.

Response 2 : Thank you for your comments. We acknowledge your concern regarding the cohort size of 41 patients. However, the sample size calculation for this study was rigorously based on the primary outcome—quality of life (QoL)—and on previous studies assessing QoL in similar patient populations. We calculated the required sample size using the formula for estimating sample size in cross-sectional studies with continuous outcomes, assuming a moderate effect size (d = 0.6), a significance level (alpha) of 0.05, and a statistical power of 0.80. Estimating the standard deviation of QoL scores to be 1.0, we calculated that approximately 39 patients would be necessary to detect a meaningful effect. Given that our final cohort included 41 patients, our sample size meets these methodological requirements, ensuring the statistical power necessary to support our conclusions.

Regarding the representativeness of the patient population, while oral squamous cell carcinoma (OSCC) is indeed a common cancer globally, the focus of our study was specifically on patients who had completed their treatment and were experiencing aesthetic and psychosocial consequences, making them a unique subset. The inclusion criteria were designed to reflect real-world clinical practice, where OSCC patients have diverse treatment histories and outcomes. The sample, though limited in size, captures a relevant and representative cohort for studying the impact of mask-wearing on QoL in this particular context. The study’s findings offer valuable insights into the psychosocial experiences of these patients, which can inform supportive care strategies.

Furthermore, our study focuses on a specific cohort: patients surgically treated for oral squamous cell carcinoma (OSCC) during the COVID-19 mask mandate. This period was marked by reduced surgical and treatment activity due to the pandemic, making this cohort particularly unique. The targeted inclusion criteria allowed us to explore the psychosocial impact of mask-wearing on a subset of patients who faced distinct challenges during this time.

Regarding the robustness of the statistical analysis, despite the modest cohort size, our statistical analyses, including multiple statistical tests (paired t test, Wilcokson signed Rank test, Welch’s t test) and a mixed-models effect regression, yielded robust and statistically significant results. Additionally, the internal consistency of the questionnaire, demonstrated by a high Cronbach’s alpha (0.931), supports the reliability of the data. These statistical results indicate that the sample size, though relatively small, was sufficient to detect meaningful differences in QoL outcomes related to mask-wearing.

 In addition, in a clinical setting, especially when dealing with a specific patient population such as OSCC survivors who have completed their treatment, it is often challenging to recruit large sample sizes. This study provides important preliminary data on the psychosocial impact of mask-wearing, which could serve as a foundation for future multicentric studies with larger cohorts. We believe that our study offers an essential first step in exploring a novel aspect of patient care, and we are confident that our findings will stimulate further research in this area.

Finally, this study offers a proof-of-concept for integrating mask-wearing into supportive care strategies for OSCC patients. While modest in size, the findings highlight a low-cost, accessible intervention with potential for immediate clinical application and broader research implications, paving the way for multicenter studies to validate these results.

We have precized all these remarks in the limitation paragraph in the discussion section (page 12, lines 342-346) : “While the modest sample size and the specific context of the COVID-19 pandemic may limit the generalizability of our findings, this study serves as a valuable proof-of-concept, demonstrating the potential of mask-wearing as a simple, low-cost intervention to enhance quality of life for OSCC patients. Future larger, multicenter studies are warranted to validate these results and further explore the integration of such measures into routine supportive care strategies for cancer patients.

For all these reasons, these results and their interpretation should be approached with extreme caution, and no definitive conclusions should be drawn.“

Reviewer 2 Report

Comments and Suggestions for Authors

The study is innovative and contribute with important new knowledge and thus merits publication. Furthermore, it relies on a well performed literature search. The design (one site cross section) is acceptable, and the authors get the best out of it. The reporting of results is good and professional, maybe a bit overly detailed; it might be concentrated a bit.

However, my only - and major indeed - concern regards the statistical tests. Specifically, ANOVA and two-sample t-tests are used to compare mask and non-mask responses. This is not feasible, as it is the same patients who are asked the mask and non-mask questions. Therefore, the two independent samples t tests should (must) be replaced with a paired samples t test; a description can be found elsewhere, for example https://libguides.library.kent.edu/spss/pairedsamplesttest.

Also, for the regression analyses, there is an assumption of independent observations that does not hold, cf. above. Rather, a regression approach controlling for individual effects (panel regression or mixed model) should apply.

My recommendation would be to consult a statistician, if the authors feel uncertain about the methods.

If these shortcomings were remedied, then I would recommend publication.

Author Response

Comment 1 : The study is innovative and contribute with important new knowledge and thus merits publication. Furthermore, it relies on a well performed literature search. The design (one site cross section) is acceptable, and the authors get the best out of it. The reporting of results is good and professional, maybe a bit overly detailed; it might be concentrated a bit.

However, my only - and major indeed - concern regards the statistical tests. Specifically, ANOVA and two-sample t-tests are used to compare mask and non-mask responses. This is not feasible, as it is the same patients who are asked the mask and non-mask questions. Therefore, the two independent samples t tests should (must) be replaced with a paired samples t test; a description can be found elsewhere, for example https://libguides.library.kent.edu/spss/pairedsamplesttest.

Also, for the regression analyses, there is an assumption of independent observations that does not hold, cf. above. Rather, a regression approach controlling for individual effects (panel regression or mixed model) should apply.

My recommendation would be to consult a statistician, if the authors feel uncertain about the methods.

If these shortcomings were remedied, then I would recommend publication.

Response 2 : Thank you very much for your comment.

You have raised valid concerns regarding the statistical methods, particularly about the paired nature of the data and the assumptions of independence in the regression analyses. We therefore called in a biostatistician (Dr Maryem Rhanaoui) and completely reworked all the tests, analyses and results (texts and graphs). This also enabled us to simplify/reduce the number of analyses.

Due to the small number of participants and the non-normal distribution of the responses to questions, a Wilcoxon Signed-Rank Test was chosen rather than a paired t test.

On the other hand, a Welch’s t-test was used to compare independent subgroups (patients with altered versus unaltered physical appearance) according to the same variable (with or without wearing a mask).

To evaluate correlations between categories, categorical variables were converted into numerical values using label encoding, and a Pearson correlation matrix was computed to measure relation between categories and response means.

To evaluate the influence of categories on responses to questions according to mask condition, a mixed-effects regression model was employed

This is described in detail in section 2.10 Statistical analysis.

Round 2

Reviewer 2 Report

Comments and Suggestions for Authors

I am very pleased to see that you invited a professional biostatistician. The revisions that you suggest are very professional, and I agree that the sign-rank test is preferable for the paired t-test. Thus, I am fully satisfied with the revision and recommends acceptance without reservation.